# Exploring techno-economic landscapes of abatement options for hard-to-electrify sectors

Clara Bachorz [1,2] ✉, Philipp C. Verpoort [1], Gunnar Luderer [1,2] & Falko Ueckerdt [1]

Approximately 20% of global $CO_2$ emissions originate from sectors often labeled as hard-to-abate, which are challenging or impossible to electrify. Alternative abatement options are necessary for these sectors but face critical bottlenecks, particularly concerning the availability and cost of low-emission hydrogen, carbon capture and storage, and non-fossil $CO_2$ for synthetic fuels or carbon-dioxide removal. In this study, we conduct a broad techno-economic analysis, mapping abatement options and hard-to-electrify sectors while addressing associated technological uncertainties. Our findings reveal a diverse mitigation landscape that can be categorized into three tiers, based on the abatement cost and technologies required. By requiring long-term climate neutrality through simple conditions, the mitigation landscape narrows substantially, with single options dominating each sector. This clarity justifies targeted political support for sector-specific abatement options, increasing investment security for transforming hard-to-electrify sectors.

While direct electrification can abate most emissions in end-use sectors, such as road transport or residential heating[1], a notable share of 20% of $CO_2$ emissions originated from sectors deemed hard-to-electrify (HTE) in 2022[2–4]. These sectors, which often overlap with the hard-to-abate category[5,6], include long-distance aviation and maritime freight transport, chemical feedstocks, and primary steel and cement production. Reducing emissions in HTE sectors will likely encounter greater challenges and slower progress, making them bottlenecks for reaching climate neutrality. According to the International Energy Agency's Net Zero by 2050 scenario, HTE sectors are projected to account for 62% of residual gross $CO_2$ emissions at the time of climate neutrality, which is in line with numerous governments' long-term projections[7]. The sectors' assets also have long lifetimes[8] of up to 30–70 years, which increases the risk of fossil lock-ins and further threatens climate targets[9]. To mitigate this risk, investments must be urgently redirected toward low-emission technologies for HTE sectors[10,11], requiring a better understanding of the most promising abatement options.

Current integrated assessment model (IAM) scenarios often rely on carbon-dioxide removal (CDR) and bioenergy to abate remaining $CO_2$ emissions from HTE sectors[12]. However, the feasibility of these options faces substantial obstacles. CDR options are either highly land-intensive, such as afforestation or bioenergy with carbon capture and storage (BECCS)[13], or immature and not being scaled up quickly enough, like direct-air capture with carbon storage (DACCS) or enhanced weathering[14]. Regarding bioenergy, constraints on land availability and concerns about biodiversity impacts restrict the sustainable utilization of this resource[15]. Furthermore, under incomplete protection of global forests, bioenergy can induce substantial indirect land-use change emissions[16]. Given these considerations, while certain forms of CDR compensation will likely be at least partially required to mitigate emissions from the HTE sectors, it is essential to explore alternative abatement routes to ensure a robust and sustainable pathway to climate neutrality.

Five abatement options are most commonly considered for the HTE sectors: low-emission hydrogen or ammonia[5], low-emission synthetic fuels (in certain cases referred to as e-fuels)[17], carbon capture

[1]Potsdam Institute for Climate Impact Research, Potsdam, Germany. [2]Global Energy Systems Analysis, Technische Universität Berlin, Berlin, Germany. ✉e-mail: clara.bachorz@pik-potsdam.de

and storage (CCS)[18], fossil carbon capture and utilization (CCU), and emissions compensation via carbon-dioxide removal (CDR), such as DACCS. However, these abatement options are currently deployed at low levels, largely due to their dependence on low-emission hydrogen ($H_2$), non-fossil $CO_2$ (of biogenic or atmospheric origin), or $CO_2$ transport and storage. These three elements are bottlenecks – components that are critical to the above-mentioned abatement options, but often characterized by high present costs, future cost uncertainty, and, in some cases, low technological maturity. These factors contribute to high investment uncertainty, rendering their ramp-up speed and future availability uncertain.

This study offers a comprehensive techno-economic assessment of five large hard-to-electrify sectors. Using a comprehensive set of techno-economic data based on 20 studies from 2013 to 2024, we analyze and compare key abatement options and derive sector-specific mitigation landscapes by varying the cost parameters of the identified bottlenecks. These landscapes indicate the least-cost abatement options and assess their robustness against uncertain developments in bottleneck parameters, such as the cost of low-emission hydrogen and the cost of non-fossil $CO_2$. This approach allows us to explore the role of individual abatement options and differentiate the abatement challenges across HTE sectors.

Building on this analysis, we also address the role of fossil CCU and examine the coordination challenges surrounding it, showing that HTE sectors cannot rely on fossil point sources once CCS is available. Furthermore, our comparison of abatement costs of low-emission synthetic fuels and CDR compensation for aviation highlights the cost-competitiveness problem of these synthetic fuels. Finally, we extend our abatement cost framework to also include residual $CO_2$ emissions of partial abatement options, demonstrating its effectiveness in identifying potential stranded assets, particularly in the steel sector.

## Results

### Alternatives to direct electrification and their bottlenecks

We undertake a comparative analysis of the five major abatement options (Options 1–5) for the HTE sectors, focusing on those with the greatest potential. Figure 1 provides an overview of Options 1–5 and a schematic of their corresponding carbon flow fingerprints.

Option 1 is the use of low-emission hydrogen ($H_2$), either directly or through the synthesis of low-emission ammonia. There are two main ways to produce low-emission $H_2$: green $H_2$, produced from water electrolysis powered by low-carbon electricity, and blue $H_2$, produced from the steam reforming of natural gas with CCS. Specifically, we focus on steam-methane reforming (SMR) with high capture rates – greater than 90% – corresponding to direct $CO_2$ emissions of less than 31 kg $CO_2$ $MWh_{H2}^{-1}$ (lower heating value) or 1 kg $CO_2$ $kg_{H2}^{-1}$. Supplementary Note 9 analyses the competition between both options, while Supplementary Fig. 6 shows the cost of different $H_2$ production processes.

Option 2 is fossil CCU, which refers to capturing fossil $CO_2$ from a point source (e.g., cement or steel plants) and using it to synthesize a new product. Currently, $CO_2$ from fossil CCU is primarily used in urea production and refinery processes. Emerging applications include synthesizing fuels and feedstocks by combining captured fossil $CO_2$ with low-emission $H_2$, which could help decarbonize the transport and chemical production sectors. This is the sole CCU application relevant to the HTE sectors and is further referred to as CCU, while the fuels produced are called *fossil CCU synfuels*. While fossil CCU synfuels reduce the combined emissions from both source and usage applications, they do not eliminate them and achieve, at best, a 50% reduction. The extent of emissions abatement for an individual application depends on how residual emissions are attributed between the two, which we refer to as the CCU attribution.

Option 3 is low-emission synthetic fuels (low-emission synfuels), which use non-fossil $CO_2$ (from atmospheric or biogenic origin) and low-emission $H_2$. When produced with electrolytic hydrogen, they are referred to as e-fuels. A wide range of fuels, such as synthetic methanol or synthetic jet fuel, can be produced this way.

Option 4 is CDR compensation, specifically through DACCS and CCS of biogenic $CO_2$ in this study. This creates negative emissions to offset the continued use of fossil fuels.

Option 5 is CCS, which refers to capturing $CO_2$ from a fossil point source, followed by its long-term underground storage, such as in a saline aquifer[19].

As of today, Options 1–5 only exist at low deployment levels. This is a result of their dependency on key bottlenecks (Fig. 1): low-emission

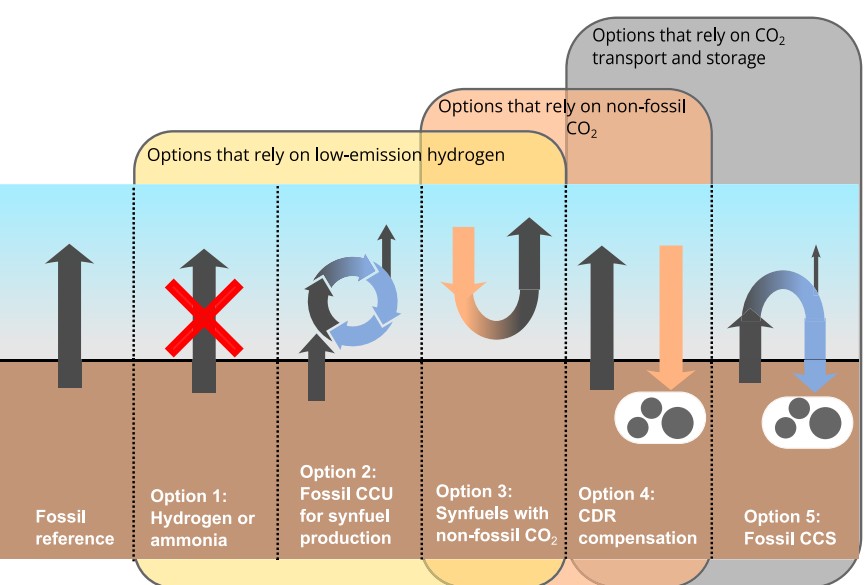

**Fig. 1 | Carbon flow fingerprints of the five categories of abatement options (Options 1–5) discussed in this work.** The arrows illustrate the flow of carbon characteristic of each option. Orange arrows represent the use of direct-air capture (DAC) or carbon capture of biogenic $CO_2$, and blue arrows the capture of point-source $CO_2$ from an industrial source. Their dependence on bottlenecks, such as low-emission hydrogen ($H_2$), non-fossil $CO_2$ (from DAC or of biogenic origin), and $CO_2$ transport and storage, is also shown using the three yellow, orange, and gray boxes. CCU carbon capture and utilization, CCS carbon capture and storage, CDR carbon-dioxide removal (via DACCS or carbon capture of biogenic $CO_2$).

$H_2$, non-fossil $CO_2$, and $CO_2$ transport and storage infrastructure, which face substantial uncertainty regarding future availability and costs.

Concerning low-emission hydrogen, numerous green $H_2$ projects have been announced, yet few have reached a final investment decision, creating substantial uncertainty about their realization[20]. Reasons for this include high costs today, high uncertainty in future cost developments, and insufficient policy backing[21]. While blue $H_2$ is expected to be more affordable than green $H_2$, at least in the short-term to mid-term, it requires high carbon capture rates to reach satisfactory $CO_2$ abatement potential[22], minimization of any methane leakage along the supply chain[23], and availability of $CO_2$ transport and storage. The current deployment and investment decisions of blue $H_2$ are similar in magnitude to those of green $H_2$[21]. Furthermore, scaling up hydrogen faces the broad systemic challenge of establishing a new energy carrier with corresponding infrastructure, supply, and demand technologies. While the main analyses in this paper compare low-emission hydrogen to other abatement options, the competition between blue and green hydrogen is further explored in Supplementary Note 9.

Non-fossil $CO_2$, whether of biogenic origin or captured via DAC, also faces availability and cost challenges. DAC is a nascent technology, with only a few pilot projects in operation[24], high uncertainty in future availability[25,26], and long-term costs[27]. The high green electricity and heat demands associated with DAC result in significant energy costs, compounded by high capital expenditure[28]. Biogenic $CO_2$ could be a lower-cost option; however, it requires capturing waste $CO_2$ from decentralized sites such as waste and biogas plants, or from pulp and paper production facilities. Most importantly, the amount of $CO_2$ available from such sources is also intrinsically limited[29]. We further explore some of these drawbacks in Supplementary Note 10 and Supplementary Fig. 8.

Finally, $CO_2$ transport and storage require pipelines or ships transporting $CO_2$, sufficient injection capacity, and geological storage availability, none of which exist at large scales today. While the largest component in CCS costs will likely be $CO_2$ capture costs rather than storage and transport[19,30], past failure rates[31] and future projections indicate short-term scarcity and long-term uncertainty[32]. Thus, although costs may be moderate, $CO_2$ transport and storage scale-up and availability remain a crucial bottleneck.

The extent to which these bottlenecks hinder abatement efforts varies across sectors, as each has distinct technological requirements and decarbonization pathways. Two of the five HTE sectors considered are transport sectors (aviation and maritime), where $CO_2$ emissions result from the combustion of fossil fuels. This study focuses on long-distance aviation and maritime transport, as shorter distances could potentially be electrified based on batteries[33,34]. Decarbonizing these sectors hinges on finding new, climate-neutral fuels to replace the existing jet fuel and heavy fuel oil (HFO), or by emission compensation using CDR.

The three remaining HTE sectors − cement manufacturing, primary steel production, and chemical feedstock usage − are heavy industry sectors. $CO_2$ emissions in cement and steel occur as a result of energy and process emissions, while chemical feedstocks have an additional contribution from end-of-life emissions, for example, when plastic waste is incinerated. As a result, these sectors need to capture $CO_2$ emissions for storage (CCS) or usage (fossil CCU), compensate for emissions with CDR, or replace fossil fuels with climate-neutral feedstocks (low-emission $H_2$ in direct reduction of iron ore or low-emission synfuels as carbonaceous chemical feedstock). Table 1 summarizes the different options considered for each sector, with more details available in Supplementary Notes 4–8.

Alternative abatement options exist for the HTE sectors that are not covered in this work. One option is using Fischer-Tropsch biofuels. Instead, we include synthetic fuels using the methanol route, with

**Table 1 | Overview of the abatement options considered in this study for each hard-to-electrify sector**

| | Fossil reference | $H_2$/$NH_3$ | Fossil CCU | Low-emission synfuels (with non-fossil $CO_2$) | Fossil reference + CDR compensation | Fossil CCS |
|---|---|---|---|---|---|---|
| Long-distance maritime | ICE + fossil HFO | ICE + Ammonia | ICE + Syn-methanol using fossil $CO_2$ from CCU | ICE + Syn-methanol using non-fossil $CO_2$ | Fossil HFO + CDR compensation | N/A |
| Long-distance aviation | Fossil jet fuel | N/A | Syn-jet fuel using fossil $CO_2$ from CCU | Syn-jet fuel using non-fossil $CO_2$ (methanol route) | Fossil jet fuel + CDR compensation | N/A |
| Cement | Cement plant | N/A | Cement with integrated calcium looping CCS, $CO_2$ available for utilization | N/A | Cement plant + CDR compensation | Cement with integrated calcium looping CCS, storage of captured $CO_2$ |
| Primary steel | Blast furnace with basic oxygen furnace (BF-BOF) | Direct reduction of iron with $H_2$ + electric arc furnace | BF-BOF + MEA CCS, captured $CO_2$ available for utilization | N/A | BF-BOF route + CDR compensation | BF-BOF + MEA CCS, storage of captured $CO_2$ |
| Chemical feedstocks (olefin production) | Naphtha cracking | N/A | Syn-methanol using fossil $CO_2$ from CCU used in the MTO process | Syn-methanol using non-fossil $CO_2$ used in the MTO process | Naphtha cracking + CDR compensation | N/A |

$H_2$ low-emission hydrogen, $NH_3$ low-emission ammonia, CCU carbon capture and utilization, CDR carbon-dioxide removal, CCS carbon capture and storage, ICE internal combustion engine, HFO heavy fuel oil, MEA monoethanolamine, MTO methanol-to-olefin.

biogenic carbon. However, in both cases, it should be noted that concerns exist regarding the sustainability of their large-scale use[15,35–37]. Another option is energy and material efficiency measures. Despite their opportunities, their potential is limited for the HTE sectors. Energy efficiency measures cannot abate process emissions (chemical feedstocks, cement), and only have a small reduction potential for the transport sectors[38]. As for steel, although major improvements have been observed since the early 20th century, the global average greenhouse gas intensity of steel production has stagnated since 1995[39]. While material efficiency and circularity measures could potentially reduce demand for cement, chemical feedstocks, and steel in some regions, the global demand for primary materials is expected to double by 2060, likely offsetting these gains[40].

## Comparing abatement costs under technological uncertainty to derive mitigation landscapes

To compare the abatement options assessed here, we conduct a techno-economic analysis and calculate the abatement costs. As previously emphasized, most of the options considered are highly sensitive to either the cost of non-fossil $CO_2$, the cost of low-emission $H_2$, or both. These cost components strongly influence the abatement cost of one option relative to another. As shown in Supplementary Fig. 1, the largest difference in levelized cost between different abatement options is given by the cost of non-fossil $CO_2$ or low-emission $H_2$.

The steel sector is a prime example of the importance of accounting for technological uncertainty when calculating abatement costs. The two most relevant abatement options, to replace the traditional blast furnace with a basic oxygen furnace (BF-BOF), are a BF-BOF plant with integrated CCS (BF-BOF-CCS) and a more transformative alternative, direct reduction of iron with $H_2$ combined with an electric arc furnace ($H_2$-DRI-EAF).

The assumed cost of $H_2$ strongly affects the levelized cost of the $H_2$-DRI-EAF option (Fig. 2) and thereby determines the difference in abatement cost of the two options[41,42]. We find that the levelized cost of $H_2$-DRI-EAF exceeds that of BF-BOF-CCS if $H_2$ costs are above €100 MWh$^{-1}$ (for a central assumption on the cost of carbon capture on a BF-BOF) (Fig. 2a). When incorporating the respective $CO_2$ emissions and calculating the abatement cost (Fig. 2b), for our central parameter assumptions, BF-BOF-CCS has a lower abatement cost for $H_2$ costs greater than €122 MWh$^{-1}$. The exact break-even cost will vary depending on the future cost of coking coal, PCI coal, and scrap steel, as well as on the CAPEX for BF-BOF-CCS and $H_2$-DRI-EAF.

We generalize the above analysis to calculate the abatement cost of all options for each sector, simultaneously assessing sensitivity to two key parameters: the cost of low-emission $H_2$ and the cost of non-fossil $CO_2$. We use $H_2$ costs ranging across €0–240 MWh$^{-1}$ and $CO_2$ costs of €0–1200 tCO$_2$$^{-1}$, which covers the high costs observed today and the low costs that might be achieved in the long term. For each unique pair of parameters, the most cost-efficient abatement option for each sector is identified. This approach assesses the robustness of a given option to future technological developments of these two key dependencies, and directly investigates future uncertainties created by the pending development of low-emission $H_2$ and non-fossil $CO_2$. The analysis is carried out for three different sets of conditions for CCU and residual emissions (Fig. 3; details in "Methods" section). Additionally, plausible parameter ranges for the cost of non-fossil $CO_2$ and low-emission $H_2$ in 2050 are shown alongside our results (gray boxes in Fig. 3). The ranges for low-emission $H_2$ are derived by combining the IEA's Net Zero Emission by 2050 (NZE) scenario for the cost of green hydrogen[43] with our calculation for the cost of low-emission blue $H_2$ (see Supplementary Note 9 on blue hydrogen for more details). The ranges for non-fossil $CO_2$ are taken from the latest updated cost estimations for DAC from Climeworks[27] and literature cost estimations for biogenic $CO_2$[44].

Without further conditions for CCU and residual emissions, the HTE sectors can be distributed into three tiers (Fig. 3a). The first tier of sectors is composed of the cement sector only, which relies primarily on CCS as an abatement option (blue). CCS remains the most competitive option (i.e., with the lowest abatement cost) if non-fossil $CO_2$ costs are above €150 tCO$_2$$^{-1}$. Below that cost, CDR compensation (i.e., offsetting fossil emissions) becomes the most competitive abatement option. The higher competitiveness of CCS relative to most CDR compensation options considered in this study is due to the technologically less challenging and more economical process of capturing $CO_2$ from flue gases with relatively high $CO_2$ concentrations (20–30%)[45] in cement production.

The second tier consists of the steel and maritime sectors, for which $H_2$-based abatement options take up a significant portion of the mitigation landscape (yellow). These options are complemented by another abatement option if low-emission $H_2$ costs are too high. For the steel sector, this complementary option is BF-BOF-CCS, and the transition to a hydrogen-based solution ($H_2$-DRI-EAF) occurs when $H_2$ costs are below €120 MWh$^{-1}$. The maritime sector depends mainly on ammonia combustion engines and, if hydrogen costs are too high, on CDR compensation. For both tiers mentioned above (cement, steel, and maritime), we find abatement costs of €50–400 tCO$_2$$^{-1}$ in 2050, with steel and cement displaying lower costs of €50–100 tCO$_2$$^{-1}$, which is low given they are considered hard-to-electrify sectors.

In contrast, the third and final tier of sectors, composed of chemical feedstocks (CF) and aviation, is more difficult to decarbonize. This is mainly due to both sectors requiring a carbonaceous fuel or feedstock to operate, which eliminates any $H_2$-only abatement option and, in the absence of CCS options, leaves only CCU, CDR compensation, or low-emission synfuels as available options. We find that CF and aviation have close-to-identical mitigation landscapes and rely on using synfuels with fossil $CO_2$ from CCU and CDR compensation, which are expensive abatement options. As a result, CF and aviation display higher abatement costs of €200–800 tCO$_2$$^{-1}$.

Fossil CCU is a predominant option for the CF and aviation sectors (gray in Fig. 3). This is a result of fossil $CO_2$ from CCU being much more affordable than non-fossil $CO_2$, with the abatement cost of low-emission synfuels always higher than that of fossil $CO_2$-based synfuels. However, the fossil CCU option is not cost-competitive for any of the potential fossil $CO_2$ source sectors, i.e., cement and steel, as CCU has a higher abatement cost than CCS (this depends on the assumed $CO_2$ transport and storage cost, a sensitivity explored in Supplementary Fig. 2). While in the short-term, other fossil $CO_2$ point sources could be used by the CF and aviation sectors, this is expected to change in the long term, as most non-HTE energy sectors transition to fossil-free alternatives. As a result, it is expected that most fossil $CO_2$ point sources other than HTE sectors will be eliminated.

Therefore, we explore a second case that requires coordination between CCU suppliers and users (Fig. 3b). Specifically, we assume the availability of fossil CCU as an abatement option to be contingent on it being the abatement option with the lowest-cost in at least one user sector and at least one supply sector for a given set of parameters (low-emission $H_2$ and non-fossil $CO_2$ costs) (see "Methods" section for implementation details). Under this additional constraint, no CCU agreement can be reached between the user and supply sectors for our base assumption of 50% CCU emission attribution to the user sectors. Consequently, fossil CCU-based synfuels disappear from the mitigation landscape of the CF and aviation sectors, as the cement and steel sectors are dominated by CCS.

For the source sectors (cement and steel), CCS is an attractive abatement option that has costs only slightly higher than those of CCU (due to additional $CO_2$ transport and storage costs) but with much lower emissions. At a CCU attribution of 50%, the cement and steel sectors have to bear half the emissions from the user sectors, on top of any residual emissions due to imperfect carbon capture rates. To

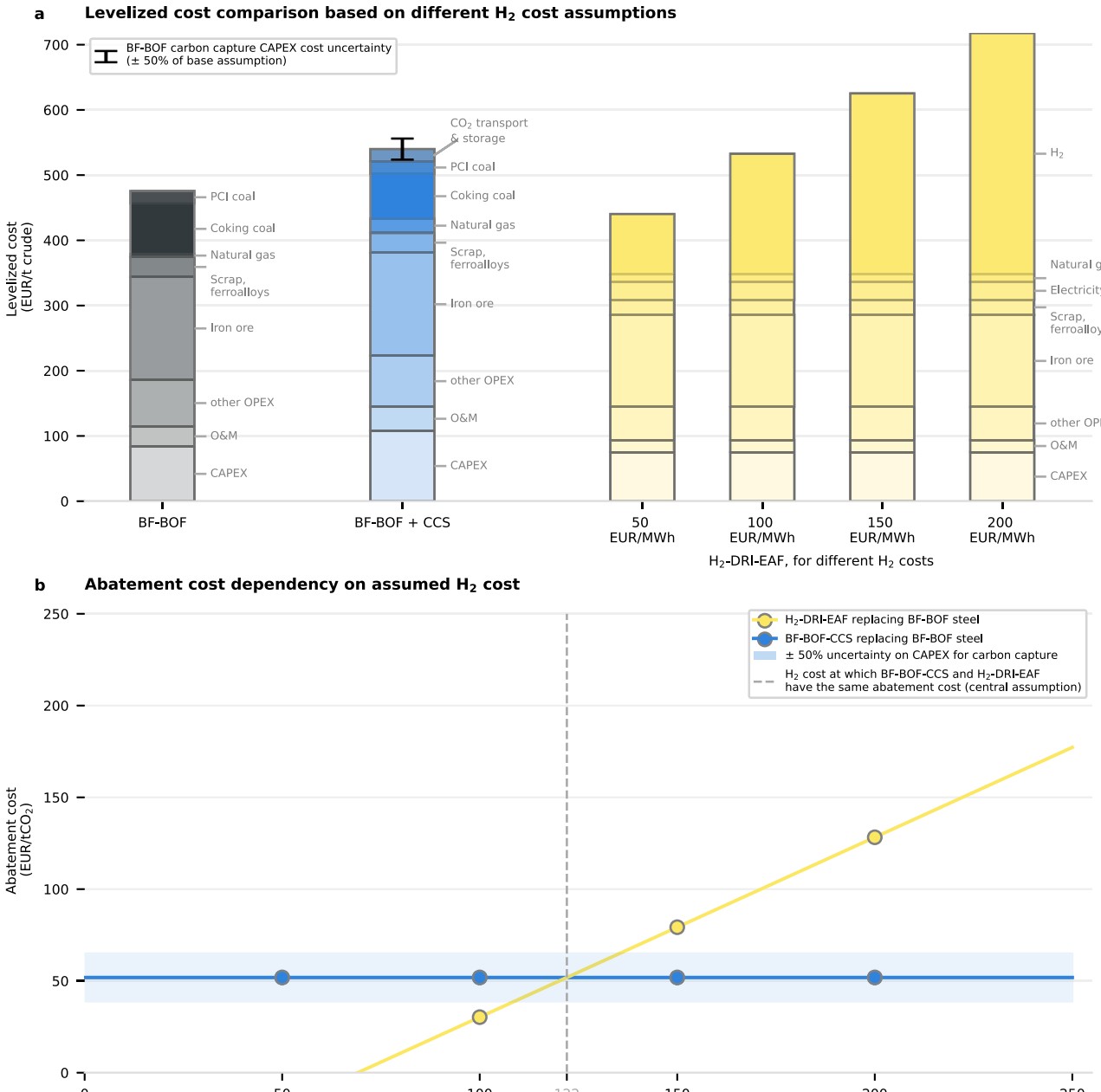

**Fig. 2 | Cost breakdown of abatement options for steel for different hydrogen cost assumptions (new plant). a** Levelized cost breakdown for the fossil steel reference, a blast furnace with a basic oxygen furnace (BF-BOF), compared with two greener alternatives, BF-BOF-CCS (BF-BOF with carbon capture and storage) and $H_2$-DRI-EAF (direct reduction of iron with an electric arc furnace operating with low-emission hydrogen). The levelized cost breakdown of the $H_2$-DRI-EAF option is shown under four assumptions for the cost of low-emission hydrogen ($H_2$). The capital expenditure (CAPEX) for carbon capture on a BF-BOF is also varied by ±50%, as indicated by the error bar (SD). **b** Calculated abatement costs for both options, as a function of the hydrogen cost assumed. The blue ribbon indicates the uncertainty on the abatement cost of BF-BOF-CCS, based on varying the BF-BOF carbon capture CAPEX. OPEX operational expenditure, O & M operation and maintenance costs, PCI pulverized coal injection.

increase the competitiveness of CCU relative to CCS, the CCU attribution would need to be shifted toward the user sector.

We investigate this question for the case of the CCU coordination between the cement and aviation sectors by varying the CCU attribution across 50–80% (Fig. 4). We observe that CCS remains the lowest-cost abatement option for the cement sector for CCU attributions of up to 70%. At a CCU attribution of 80%, CCU becomes competitive for the cement sector, but is already too expensive for the aviation sector, which instead turns to CDR compensation. Additionally, we highlight that the non-fossil $CO_2$ and $H_2$ costs assumed here are €800 $tCO_2^{-1}$ and €50 $MWh^{-1}$, which should favor

the CCU option by driving up the cost of CDR compensation for aviation and reducing the cost of synfuels.

This demonstrates that, for the $H_2$ and $CO_2$ parameters used, no CCU window exists — that is, a CCU attribution at which the aviation and cement sectors both have CCU as their lowest-cost option. Consequently, while fossil CCU is a cost-effective abatement option for HTE user sectors such as aviation or CF, finding a corresponding emission attribution that also benefits the source sector is challenging, if not impossible in some cases. As a result, if cement and steel are the only remaining possible fossil CCU source sectors by 2050, we find that a CCU agreement would be difficult to reach between the

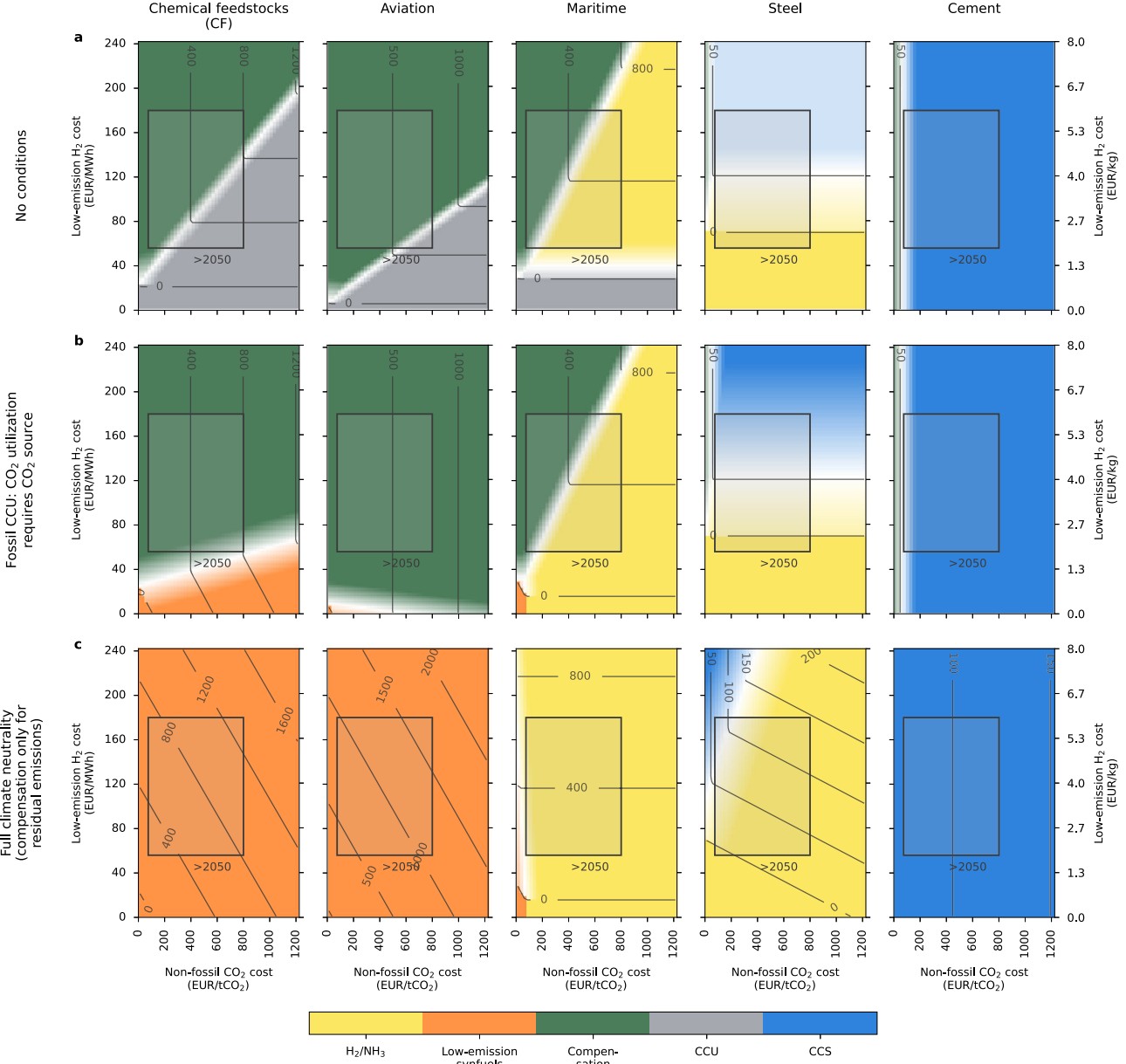

**Fig. 3 | Mitigation landscape for the hard-to-electrify sectors showing technology options with the lowest abatement cost dependent on the low-emission hydrogen and non-fossil CO₂ costs, for three cases. a** Case with no conditions. **b** Case where fossil CO₂ from CCU (carbon capture and utilization) must be sourced from the supplier sectors (steel and cement). **c** Case combining the previous requirement, and additionally only allowing compensation of residual emissions, excluding the option of compensating otherwise unabated fossil emissions (CDR), and imposing climate neutrality. The "Methods" section provides further details on how the three panels are derived. The gray boxes illustrate low-emission hydrogen ($H_2$) and direct-air capture (DAC) costs for 2050 and beyond (for $H_2$, following the IEA's Net Zero Emission by 2050 (NZE) scenario[43] and our calculation for the cost of blue $H_2$ — see Supplementary Note 9 —, and for non-fossil CO₂, the latest Clime-works estimations[27] and literature values for the cost of biogenic carbon[44]). The gray contours with numbers indicate the abatement cost contours. The linear white shading indicates when the abatement cost difference between the lowest-cost and second-lowest-cost abatement option is €100 tCO₂⁻¹ or less. CCS Carbon capture and storage, CCU carbon capture and utilization, NH₃ low-emission ammonia.

HTE sectors. We note an exception at around 85% attribution, where a small but unstable CCU window opens, which disappears when requiring climate neutrality (see Supplementary Fig. 9 for more details).

Additionally, it is worth noting that the exclusion of CCU as a viable abatement option for the HTE sectors depends on the availability of CCS as an alternative, particularly for the cement sector. In the absence of geological carbon storage, the cement sector has no alternative but to turn to CCU, as neither CCS of cement emissions nor CDR compensation would be possible. However, fossil CCU inherently leads to significant residual emissions. Achieving a climate-neutral

cement sector therefore depends on the development of the technologies that are prerequisites for CCS − including geological carbon storage and its associated infrastructure.

As fossil CO₂ from CCU becomes unavailable, CF and aviation switch from fossil CCU synfuels to low-emission synfuels (Fig. 3b). However, the higher abatement cost of these fuels results in the CDR compensation option taking up a larger area of the mitigation landscape compared to when CCU is unconstrained (Fig. 3a). This demonstrates that when low-emission synfuels and CDR compensation compete, the latter is almost always the most cost-effective solution, especially for $H_2$ costs above €40−50 MWh⁻¹.

The reduced competitiveness of low-emission synfuels compared to CDR can be better understood by analyzing their cost components. Both options share two key cost components: the cost of using fossil assets and infrastructure and the cost of non-fossil $CO_2$. The former is equal only if synfuels are drop-in replacements that require no demand-side changes (e.g., synthetic jet fuel in aviation, see Fig. 5). The latter is equal only when accounting for the $CO_2$ losses due to imperfect conversion during the synfuel synthesis process, which increase the amount of $CO_2$ required for synfuel synthesis relative to CDR compensation.

Other cost components are exclusive to either CDR compensation or low-emission synfuels. For CDR compensation, these are the cost of fossil jet fuel, $CO_2$ transport, and $CO_2$ storage. In contrast, low-emission synfuels require additional capital expenditure (CAPEX) and operational expenditure (OPEX), due to the two-stage synthesis process, involving methanol synthesis and subsequent methanol-to-jet fuel conversion. A critical cost component is low-emission $H_2$, which is the dominating component in the OPEX for methanol synthesis. Even with hydrogen costs reduced to €50 MWh$^{-1}$, low-emission synfuels for aviation have a higher levelized cost (and, since both options abate the same amount of emissions, a higher abatement cost) than CDR compensation. Therefore, the primary drivers of the cost difference and uncertainty between these two mitigation methods are the relative costs of fossil fuel and $CO_2$ transport and storage, compared to the cost of low-emission hydrogen.

The high abatement cost of low-emission synfuels relative to CDR compensation presents a significant barrier to their adoption. From a techno-economic perspective, this result suggests that political support for low-emission synfuels through subsidies or quotas may be ill-advised. However, there could be reasons to favor the low-emission synfuels option over the CDR compensation option from a societal and broader economic perspective.

First, the rate at which $CO_2$ transport and storage infrastructure can be scaled up — a prerequisite to CDR compensation options such as DACCS — is uncertain[46] and tends to be overestimated in modeling studies[32]. Given that huge amounts of CDR compensation will be required for non-$CO_2$ greenhouse gas emissions, such as methane or NOx emissions in agriculture that have no alternative abatement options, CDR may remain scarce even in the long term[14,47]. Therefore, one could argue that any CDR capacity should be prioritized for these non-$CO_2$ sectors. Relying on future CDR availability for HTE sectors such as aviation instead risks deepening the lock-in of uncompensated fossil emissions. Second, recent work[48] suggests that the cost advantage of DACCS compensation diminishes when considering non-$CO_2$ effects, as certain low-emission synfuels production processes (e.g., the Fischer-Tropsch route) reduce aromatic content and, consequently, contrail formation.

Considering these factors, while CDR compensation may appear to be an inexpensive option from a techno-economic point of view, it is accompanied by many uncertainties and caveats. To account for this, we further build on Fig. 3b and exclude the full compensation option from the mitigation portfolio available (see "Methods" section for more details).

Additionally, Fig. 3b has the drawback of comparing abatement options that do not provide the same level of emissions abatement. For instance, CCS has low abatement costs of €30–70 tCO$_2^{-1}$ [49], but only reduces emissions by 60–90% (depending on application) and leaves the remainder unabated. To compare all abatement options on the same basis, we impose compatibility with climate neutrality as an additional condition (Fig. 3c). We impose this condition by adding the cost of CDR compensation for residual emissions (see "Methods" section) to the cost of each abatement option. Therefore, we allocate substantially less total CDR capacity to the HTE sectors by reserving it only to compensate for any residual $CO_2$ emissions. This ensures that all the abatement options considered are assessed by the same criteria.

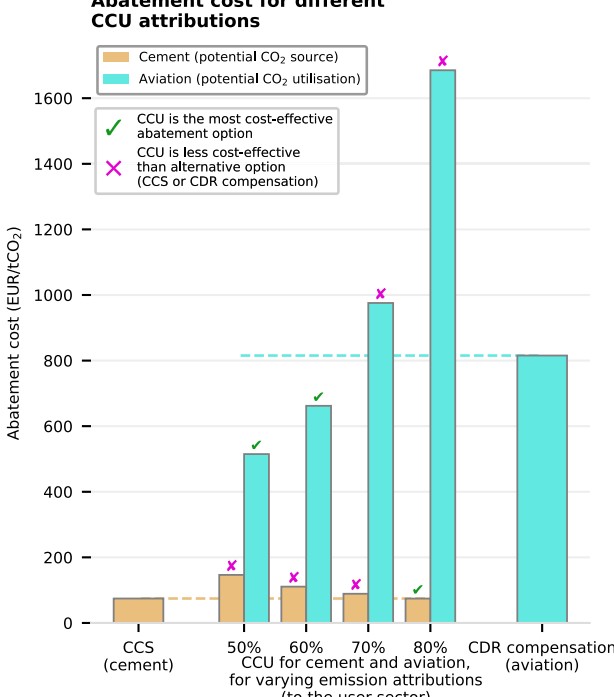

**Abatement cost for different CCU attributions**

Fig. 4 | Abatement costs for the cement and aviation sectors, with varying carbon capture and utilization (CCU) emissions attributions. The abatement options for cement are shown in orange, with CCS (carbon capture and storage) being the lowest-cost alternative to CCU. For aviation, abatement options are shown in light blue, with the most cost-effective alternative available being compensation using DACCS (direct-air capture with CCS) or CCS of biogenic $CO_2$. When the CCU option for a given attribution is more cost-effective than the alternative, a green tick is shown, and otherwise, a pink cross. The cost of non-fossil $CO_2$ is assumed to be €800 tCO$_2^{-1}$, and the cost of low-emission hydrogen is €50 MWh$^{-1}$.

With no full CDR compensation available, the CF and aviation sectors rely only on low-emission synfuels and are the most expensive HTE sectors to decarbonize. This result is in agreement with other works on aviation[50,51] and CFs[52,53], which also found low-emission synfuels to be the best abatement option for these sectors. Abatement costs of €300–1000 tCO$_2^{-1}$ and a dependency on two key bottlenecks, low-emission $H_2$ and non-fossil $CO_2$, result in CF and aviation being prime examples of hard-to-abate sectors. This can be contrasted with the maritime sector, which turns entirely to ammonia shipping and is therefore only reliant on the development of $H_2$. Low-emission syn-methanol is not found in the mitigation space of the maritime sector, a conclusion that contrasts with some previous studies, which found low-emission syn-methanol (synthesized with biogenic $CO_2$) to be the lowest-cost abatement option for most[54] or all[55] sensitivities explored. This result further illustrates the competitiveness problem of low-emission synfuels in sectors that have another direct electrification and/or hydrogen option without carbon, which has already been observed in the light and heavy road transport sectors[51,56]. However, a significant downside of using ammonia instead of methanol in internal combustion engines (ICE) is the potential for $N_2O$, $NO_x$, or $NH_3$ emissions. This could disturb the nitrogen cycle and diminish the mitigation potential of maritime transport based on ammonia combustion[57–59].

Finally, the climate neutrality requirement strongly reduces the competitiveness of BF-BOF-CCS in the steel sector. This is due to its significant residual $CO_2$ emissions of 0.7 tCO$_2$ per tonne hot-rolled coil (corresponding to a 65% $CO_2$ capture rate, see Supplementary Note 6 and Supplementary Table 6), which have to be compensated for using CDR compensation, thereby increasing abatement costs. As a result,

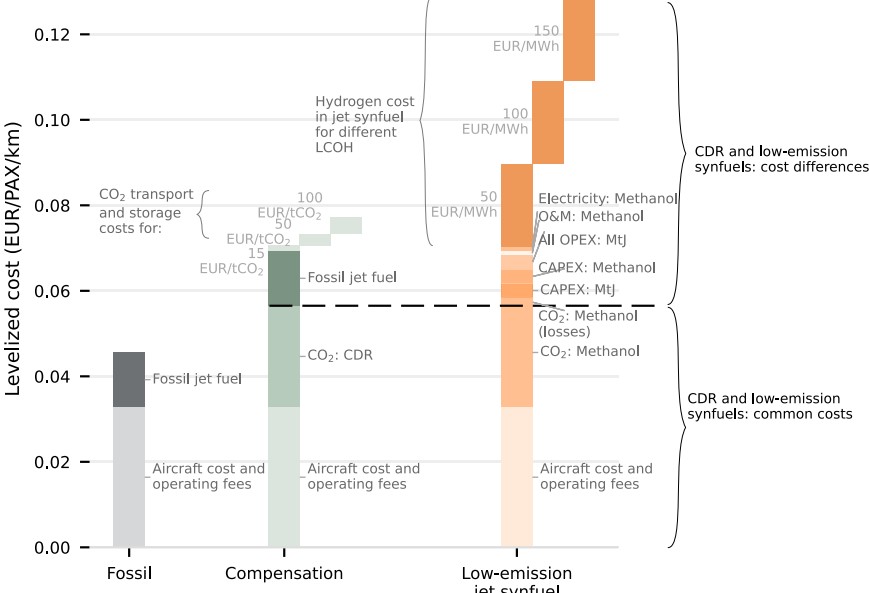

**Fig. 5 | Levelized cost comparison of low-emission synfuels and CDR compensation for the aviation sector.** The gray stacked bars illustrate the fossil fuel pathway (use of fossil jet fuel), the green bars the use of carbon-dioxide removal (CDR) compensation, and the orange bars the use of drop-in low-emission synthetic jet fuel. The light green bars that are offset show the cost component of $CO_2$ transport and storage for three different cost assumptions. The orange bars that are offset show the cost component of hydrogen (for the methanol synthesis process) under different levelized cost of hydrogen (LCOH) assumptions. The black curly braces indicate the common cost components between CDR compensation and low-emission synfuels (bottom), and the cost differences (top). The dashed line indicates the transition between both cost categories. PAX person, MtJ Methanol-to-jet fuel synthesis, O & M operation and maintenance costs.

$H_2$-DRI-EAF becomes the most cost-effective abatement option for primary steel-making under most low-emission $H_2$ and non-fossil $CO_2$ cost assumptions. This demonstrates that imposing climate neutrality limits the competitiveness of CCS, especially for sectors that have alternative abatement options. The notable exception is the cement sector, due to a much higher capture rate of 90% (as cement flue gases have much higher $CO_2$ concentrations than steel), and a lack of alternative abatement options for process emissions. Other studies have examined CC(U)S in detail for the cement sector[60,61], highlighting it as essential for advanced emission abatement.

### The impact of retrofits and residual emissions

A notable advantage of CCS as an abatement option in industry sectors is the possibility of retrofitting existing plants (in the case of post-combustion CCS). As industrial assets can have long lifetimes – typically 40 years for the steel sector – CCS retrofits have been suggested as solutions for minimizing stranded assets and locked-in emissions[62,63]. To evaluate this, we study the case of an operating BF-BOF steel plant by using a reference BF-BOF plant that has already been fully depreciated (no capital expenditure in the levelized cost components). This includes the assumption that a blast furnace relining has just been carried out (considering a typical relining cycle has a duration of 15–20 years[10,63]). We then evaluate the impact of retrofitting a BF-BOF with CC, and compare the abatement cost of the BF-BOF-CCS (retrofit and new plant) and $H_2$-DRI-EAF routes (Fig. 6). The marginal abatement cost curves include CDR compensation to offset residual emissions, ensuring climate neutrality (Fig. 6a, b).

In the case of BF-BOF-CCS, 63% of emissions can be abated at relatively low costs of €80 $tCO_2^{-1}$ (retrofit) and €160 $tCO_2^{-1}$ (new plant). However, the remaining 37% of emissions would require expensive CDR compensation. With CDR compensation costs of €365 $tCO_2^{-1}$, the abatement cost of the CCS options increases by €140 $tCO_2^{-1}$ (from €50

$tCO_2^{-1}$ to around €190 $tCO_2^{-1}$ in the retrofit case). In contrast, while $H_2$-DRI-EAF is more expensive than CCS (€120–210 $tCO_2^{-1}$, depending on the cost of $H_2$), it requires much less CDR compensation to reach climate neutrality, with the abatement cost only increasing by around €20 $tCO_2^{-1}$.

Next, we compare the combined abatement cost for both incomplete mitigation (dashed lines) and full mitigation using CDR compensation (solid lines) (Fig. 6d). We define the *break-even $H_2$ cost* (the cost at which BF-BOF-CCS and $H_2$-DRI-EAF reach the same abatement cost) and indicate it for different cases (gray squares and triangles). For incomplete mitigation, retrofitted BF-BOF-CCS (light blue, dashed line) is more cost-effective than $H_2$-DRI-EAF (yellow dashed line) up to a break-even $H_2$ cost of €50 $MWh^{-1}$ (lower gray square). However, when moving on to full mitigation, BF-BOF-CCS competitiveness suffers, with the retrofit case breaking even with $H_2$-DRI-EAF at $H_2$ costs of €150 $MWh^{-1}$ (lower gray triangle). With a newly built BF-BOF-CCS plant, the break-even $H_2$ cost further increases to €230 $MWh^{-1}$ (upper gray triangle). These results suggest that while retrofitting CCS can be an economical abatement option in the short term, achieving full mitigation in the steel sector is often more cost-effective with newly built $H_2$-DRI-EAF plants.

The exact break-even $H_2$ cost, however, depends on the cost of non-fossil $CO_2$ (as shown in Supplementary Fig. 3), as well as on the $CO_2$ transport and storage cost assumed. Additionally, we highlight that this result does not entirely rule out the use of post-combustion CCS for the steel sector. DRI-EAF with natural gas and CCS is another option that has been gaining prominence and can be an alternative option for regions with a low-cost natural gas supply if the cost of low-emission $H_2$ remains too high.

### Limitations and further work

Several limitations should be noted that point to future research directions. First, the techno-economic analysis focuses on key

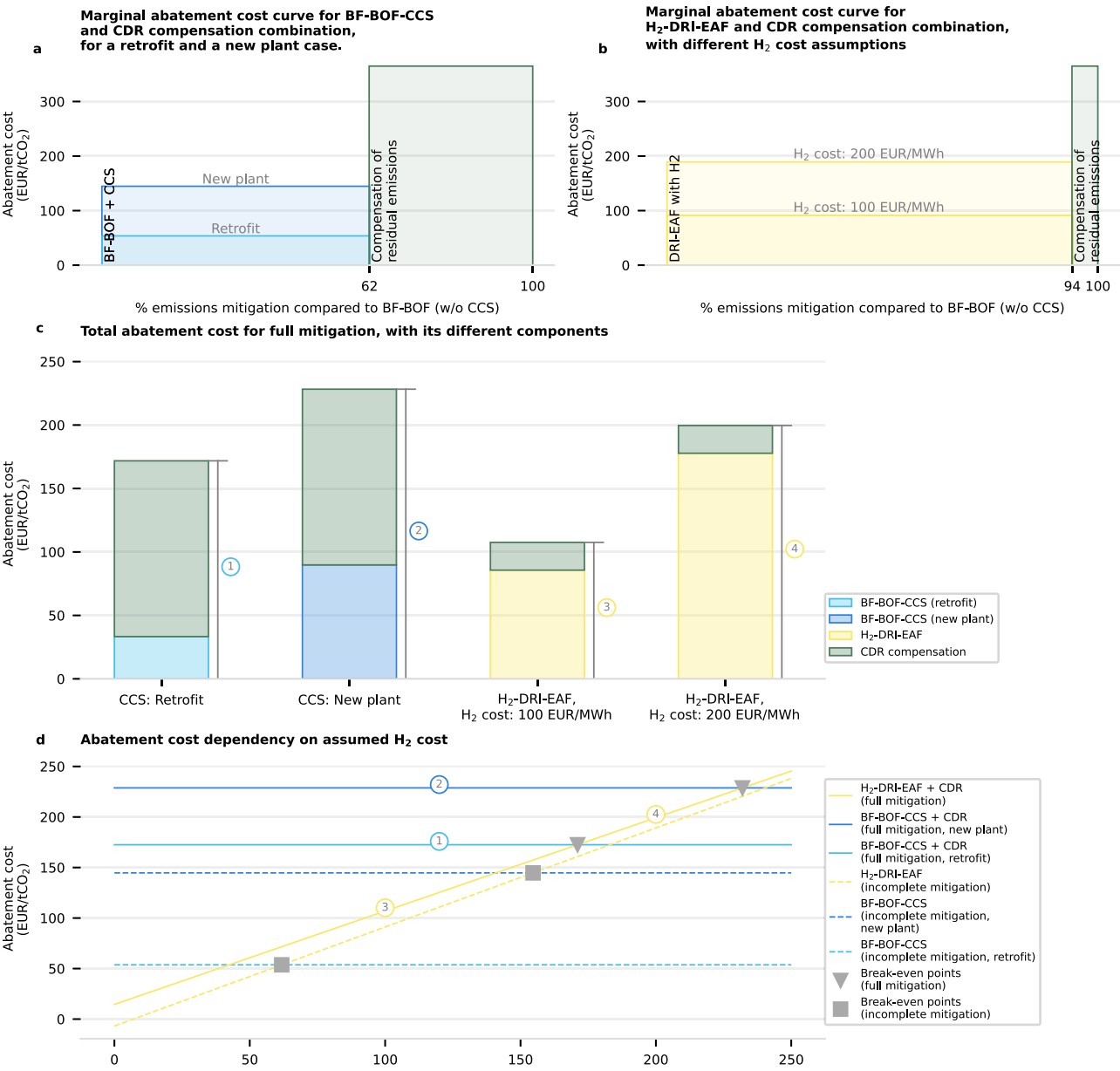

**Fig. 6 | Marginal abatement cost curve and abatement costs for different climate-neutral steel pathways relative to an existing BF-BOF plant, dependent on low-emission hydrogen ($H_2$) cost and retrofit possibility. a** Marginal abatement cost curve for climate-neutral steel, using a blast furnace with a basic oxygen furnace (BF-BOF) fitted with a carbon capture plant (BF-BOF-CCS) **b** Marginal abatement cost curve for climate-neutral steel, using direct reduction of iron with hydrogen ($H_2$) and electric arc furnace plant ($H_2$-DRI-EAF). Both options require some CDR (carbon-dioxide removal) emission compensation to be climate-neutral, which we assume to be available at €365 $tCO_2^{-1}$. **c** Total abatement cost for all four options, for the full mitigation case. The different shaded areas indicate the abatement cost portion taken up by each abatement option. **d** Abatement cost dependency on $H_2$ costs, for the full mitigation case (with compensation, full lines) and the incomplete mitigation case (no compensation, dashed lines). For the BF-BOF-CCS route, both a retrofit and a new plant case are calculated. The break-even $H_2$ cost between the BF-BOF-CCS (retrofit and full plant) and $H_2$-DRI-EAF options is also shown, for incomplete mitigation (gray squares) and full mitigation (gray triangles). The corresponding total costs from (**c**) are also shown, labeled by numbers 1–4.

abatement options and does not include all possibilities. Future studies could include less mature options, such as direct solar fuels[64,65], and demand-side and circularity measures (such as material substitution or mechanical and chemical recycling). Second, our study does not address temporal scaling requirements, endogenous technological progress, and cross-sectoral competition of abatement options. Future work could combine techno-economic assessments, which can analyze the competitiveness of technologies across broad ranges of uncertainty, with scenario analyses and energy-system modeling that captures system interactions and transition dynamics. Another source

of uncertainty identified in our study is the capital cost of BF-BOF-CCS and exact retrofit costs, although we find that this has a limited impact on final levelized and abatement costs (see Supplementary Fig. 4). Narrowing the plausible parameter space could still help determine the exact break-even cost of $H_2$, which is a key indicator of the competitiveness between CCS- and $H_2$-based steel. Finally, the absolute abatement cost of all options is dependent on the cost trajectory of fossil fuels, which is highly uncertain. Studying various scenarios for fossil-cost trajectories and their impact on HTE abatement costs could help quantify this impact.

## Discussion

As electrification expands in end-use sectors, such as road transport, buildings, and industrial heat, it will encounter fundamental barriers in sectors that are hard to abate. These hard-to-electrify (HTE) sectors include basic material production, such as cement, steel, and basic chemicals, and long-distance transport like maritime and aviation.

The main abatement options for HTE sectors rely on overcoming characteristic bottlenecks related to $CO_2$ transport and storage, low-emission $H_2$, and non-fossil $CO_2$ for producing low-emission synfuels or negative emissions. In the case of $CO_2$ transport and storage, the main challenge is large-scale availability, while low-emission $H_2$ and non-fossil $CO_2$ face issues with both large-scale availability and costs.

Here, by combining key cost uncertainties, we derive mitigation landscapes that specify the role of different abatement options and differentiate the hard-to-abate property across HTE sectors. We find that these sectors can be grouped into three tiers:

The first tier is composed only of cement production. For this sector, CCS is the dominant option, and emissions reductions are expected to be possible at low abatement costs of €50–150 $tCO_2^{-1}$ if large-scale CCS is available. Hence, climate mitigation in this sector hinges on resolving CCS scale-up bottlenecks such as societal acceptance, regulatory and legal issues, infrastructure financing, and planning and construction.

In the second tier, the steel and maritime sectors mostly rely on $H_2$ and $NH_3$, leading to moderate abatement costs of €50–400 $tCO_2^{-1}$. For the steel sector, an alternative abatement option is BF-BOF-CCS. This option displays low abatement costs, which break even with those of $H_2$-based steel at an $H_2$ cost of €120 $MWh^{-1}$ (€4 $kg^{-1}$) for the greenfield case. However, the CCS option only reduces around 60% of emissions relative to the fossil baseline. When these residual emissions are compensated for, the break-even $H_2$ cost increases to over €200 $MWh^{-1}$ (for CDR emission compensation costs of €300 $tCO_2^{-1}$ or more). For maritime transport, low-emission synthetic methanol is an alternative abatement option that eliminates the risks associated with ammonia combustion but has a substantially higher abatement cost.

In contrast, the sectors in the third tier, aviation and chemical feedstocks, typically face high abatement costs of €300–1200 $tCO_2^{-1}$. As they rely on carbonaceous fuels or feedstocks, these sectors require CDR compensation or low-emission synfuels, the most expensive abatement options. Additionally, the development of both abatement options hinges on overcoming multiple bottlenecks. CDR compensation, such as DACCS, requires both the availability of $CO_2$ transport and storage and substantial amounts of non-fossil $CO_2$, while low-emission synfuels require both non-fossil $CO_2$ and low-emission $H_2$.

We have also shown how much the mitigation landscape narrows when adding simple conditions that align the techno-economic analysis to long-term needs for climate neutrality. From these conditions, we derive the following key insights.

First, there is no substantial window for fossil CCU for the HTE sectors. While fossil CCU is incompatible with climate neutrality, it might play a role in the transitional phase. However, requiring coordination between $CO_2$ source and $CO_2$ utilization applications, while considering their respective climate-neutral alternative abatement options, shows that there is no stable window for fossil CCU. We find limited mutual incentives for cooperation for CCU between a fossil carbon source and user, as either of the two (depending on the attribution of residual emissions) tends to have an incentive to transition to alternative full abatement options instead.

Moreover, our results demonstrate that accounting for residual emissions supports full abatement options. Partial abatement options lose their cost-competitiveness advantage over full abatement options when imposing CDR compensation for residual emissions. This can be observed for blue hydrogen when methane leakage is not minimized (Supplementary Fig. 7), fossil CCU (Supplementary Fig. 9), and BF-BOF-CCS steel (for both greenfield and retrofit cases, see Figs. 3, 6). For the

steel sector, compensating residual emissions improves the competitiveness of $H_2$-DRI-EAF steel, shifting the break-even $H_2$ cost in the greenfield case from €120 $MWh^{-1}$ (€4 $kg^{-1}$) to up to €240 $MWh^{-1}$ (€8 $kg^{-1}$), depending on the cost of CDR emission compensation. This is also observed when comparing a retrofitted BF-BOF plant with CCS to a newly built $H_2$-DRI-EAF plant, which has important consequences for asset holders − in the long term, the most cost-effective abatement option will transition to new $H_2$-DRI-EAF plants, rather than CCS retrofits.

Lastly, we highlight that low-emission synfuels face fundamental competitiveness disadvantages compared to CDR compensation. These synfuels only become viable when the CDR compensation option is excluded from the mitigation landscapes, as they are generally more expensive due to the additional costs of low-emission hydrogen and fuel synthesis. However, CDR compensation, such as DACCS, requires resolving two bottlenecks: sourcing of non-fossil $CO_2$ and $CO_2$ transport and storage. This raises large uncertainties about the future availability of CDR compensation[26,46]. As CDR is required for hard-to-abate non-$CO_2$ emissions, such as methane and nitrous oxide from agriculture, it should be prioritized for these indispensable use cases. Therefore, energy and industry $CO_2$ emissions should be abated rather than compensated[66] to minimize fossil lock-in risks if CDR remains scarce.

In conclusion, we find a clear mapping of options and sectors with a dedicated dominant option for each HTE sector. Therefore, policies should aim at resolving bottlenecks by fostering innovation and scale-up of low-emission $H_2$, non-fossil $CO_2$, and $CO_2$ transport and storage, while respecting a sectoral prioritization. A guiding hierarchy includes focusing CDR compensation on non-$CO_2$ emissions in agriculture, fostering carbonaceous fuels and feedstocks for aviation, chemicals, and potentially maritime, while prioritizing CCS for mainly cement and supporting a transition to $H_2$-based steel. Overall, climate change mitigation is achievable at moderate average costs, while some sectors still face high abatement costs of above €300 $tCO_2^{-1}$ despite innovations, such that high carbon prices (or equivalent regulation) remain necessary in the long term.

## Methods

### Techno-economic analysis and data

In this study, we conduct a techno-economic analysis focusing on the most promising abatement options for the hard-to-electrify sectors. To be considered, abatement options must abate at least 50% of emissions of the fossil reference and be at a sufficient technological maturity (with a TRL equal to or above 5)[67].

The data used for the techno-economic analysis was obtained from a literature review of 20 different studies[4,19,27,42,55,68–82], and the POSTED database[83]. These sources encompass fuel synthesis processes: hydrogen liquefaction, ammonia synthesis using nitrogen from an air-separation unit, power-to-methanol (using non-fossil $CO_2$ or fossil $CO_2$ from CCU), and methanol-to-jet fuel (using non-fossil $CO_2$ or fossil $CO_2$ from CCU) (see Supplementary Tables 1, 2, and Supplementary Notes 1–3). Additionally, we collect data for the following industrial processes or transport modes: for aviation, on jet fuel and liquified-hydrogen aircraft, given in Supplementary Tables 3, 4, and explained in Supplementary Note 4. For maritime transport, we assume heavy fuel oil internal combustion engine (ICE) four-stroke ships, methanol ICE four-stroke ships, and ammonia ICE four-stroke ships (Supplementary Table 5 and Supplementary Note 5). The primary steel sector analysis focuses on blast furnaces with a basic oxygen furnace (with and without integrated amine-based $CO_2$ capture) and direct reduction of iron with $H_2$ and further processing of the reduced iron in an electric arc furnace (see Supplementary Table 6 and Supplementary Note 6), and cement considers a typical cement plant (with and without integrated calcium looping $CO_2$ capture, see Supplementary Table 7 and Supplementary Note 7). Finally, for the chemical

**Table 2 | Overview of the key techno-economic assumptions taken in this study**

| Component | Cost | $CO_2$ emissions | Comment |
|---|---|---|---|
| Electricity | €40 MWh$^{-1}$ | 0 kg $CO_2$ MWh$^{-1}$ | Assumed to be renewable electricity from flexible generation. |
| Heat | €30 MWh$^{-1}$ | 0 kg $CO_2$ MWh$^{-1}$ | Low-temperature heat (50–150 °C) provided by a heat pump running on renewable electricity. |
| Fossil $CO_2$ from CCU | €0 t$CO_2^{-1}$ | Dependent on the CCU attribution | By default, emissions are shared with 50% CCU attribution for the source and user sectors. This parameter is varied in Fig. 4. |

CCU carbon capture and utilization.

feedstocks sector analysis, we consider a naphtha cracking plant for olefin production and a methanol-to-olefin plant (see Supplementary Table 8 and Supplementary Note 8). $CO_2$ transport and storage costs, which are used for the calculation of the cost of CDR compensation, are €15 t$CO_2^{-1}$, based on the values found in ref. 19. Additionally, the parameters shown in Table 2 are taken as assumptions. The cost basis used in this study is €2020, with all currency conversions indicated, when relevant, in Supplementary Tables 1–10.

### Levelized cost of a product
The levelized cost of a product (LCOX) is given by Eq. (1):

$$LCOX = \frac{CAPEX \times ANF + OPEX}{cap.\ fac.} + other\ \ OPEX + \sum_i energy\ demand_i \times energy\ cost_i, \quad (1)$$

where the CAPEX stands for capital expenditure, the OPEX for operational expenditure, ANF for the annuity factor, and cap. fac. for the capacity factor. The last sum part of the equation encompasses any energy demand required by the asset, for example, electricity or coking coal.

The ANF is calculated using Eq. (2):

$$ANF = \frac{WACC \times (1 + WACC)^N}{(1 + WACC)^N - 1}, \quad (2)$$

where the WACC is the weighted average cost of capital, and N is the book lifetime of the asset.

The WACC, lifetime, and capacity factor for each product are mostly process-specific and obtained from the literature review conducted on techno-economic data. If these parameters were not found in previous studies, a base assumption is used: a WACC of 10%, a lifetime of 20 years, and a capacity factor of 90%. This base assumption is applied to all synthesis processes of synthetic fuels (methanol synthesis, Haber-Bosch, and Methanol-to-Jet fuel).

In this work, we first calculate the levelized cost of intermediate fuels, such as liquified low-emission $H_2$, low-emission ammonia, or carbonaceous synthetic fuels (fossil-based or non-fossil-based). Based on these LCOXs, we then calculate the cost of five specific products: a tonne of olefin, a tonne of primary hot-rolled coil steel, a tonne of cement, a MWh of fuel for the maritime sector, and a PAX (person) km$^{-1}$ for the aviation sector.

The costs of a MWh at lower heating value (LHV) of low-emission hydrogen and a tonne of non-fossil $CO_2$ are assumed to be major sensitivities in our LCOX calculation and are left as external parameters. Generally, the energy content of all fuels throughout this study is stated in LHV.

### $CO_2$ emissions accounting
Energy-related $CO_2$ emissions, as a result of the use of fossil fuels, are the largest source of $CO_2$ emissions for the five HTE sectors. We use the process-specific energy demand for natural gas, coal (bituminous, coking, pulverized coal injection), heavy fuel oil, jet fuel and naphtha, and use the emission factors taken from the IPCC 2006 Guidelines for

National Greenhouse Gas Inventories[84] (which was revised in 2019, but the values considered in this work are still up-to-date) to calculate corresponding $CO_2$ emissions.

Other than energy $CO_2$ emissions, cement production has additional process emissions from calcination, which are taken directly from De Lena et al.[77]. For olefin production, resulting $CO_2$ emissions encompass both process and end-of-life emissions and are directly calculated (for more details, see Supplementary Note 8).

### Abatement cost calculation
Using the LCOX and corresponding $CO_2$ emissions, the abatement cost is calculated using Eq. (3):

$$Abatement\ \ cost = \frac{LCOX_{green} - LCOX_{fossil}}{em_{fossil} - em_{green}}, \quad (3)$$

where the subscript fossil refers to the reference fossil technology, and the subscript green refers to the alternative low-emission technology. em refers to $CO_2$ emissions associated with either technology.

We also consider the case where abatement options are compared based on ($CO_2$) climate neutrality. In this case, the revised abatement cost (Eq. (4)) changes to incorporate the cost of compensating for any $CO_2$ emissions arising from the greener alternative:

$$Abatement\ cost = \frac{(LCOX_{green} + em_{green} \times LCOC) - LCOX_{fossil}}{em_{fossil}}, \quad (4)$$

where LCOC is the levelized cost of CDR compensation (which includes the cost of capture of non-fossil $CO_2$ and the cost of transport and storage).

### Code description and deriving the mitigation landscapes
The software used calculates the levelized costs of all possible products for each unique pair of inputs (x, y): (levelized cost of low-emission hydrogen, levelized cost of non-fossil $CO_2$). First, the synthetic fuel levelized costs (ammonia, methanol, jet fuel) are calculated before being used to determine the levelized costs for the five hard-to-electrify sectors. The three panels in Fig. 3 are then calculated using three different logical conditions.

For Fig. 3a., No conditions (base case): For each sector, using the fossil reference highlighted in Table 1, the basic abatement cost is calculated for each option. The option with the lowest abatement cost is chosen and displayed on the mitigation landscape.

For Fig. 3b, CCU coupling: The procedure from Fig. 3a is repeated with the addition of the condition of CCU coupling. The abatement cost results across sectors are compared, with the requirement that if a CCU user sector (aviation, chemical feedstocks, or maritime) has CCU as its option with the lowest abatement cost, it can choose this option if a CCU supply sector (steel or cement) also has CCU as its option with lowest abatement cost – and inversely. If this is not the case, the sector that had CCU as its option with the lowest abatement cost moves to the second-lowest option. Example 1: for a pair of parameters, aviation and chemical feedstocks both have CCU as their option with the lowest abatement cost, but steel and cement both have CCS. There is no CCU supply sector available,

so the condition is not fulfilled. Consequently, aviation and chemical feedstocks choose their second-lowest-cost abatement option instead (CDR compensation or low-emission synfuels). Example 2: for a pair of parameters, aviation, chemical feedstocks, and maritime all have low-emission synfuels as their most cost-efficient option, whilst cement has CCU. The condition is not fulfilled, since there are no CCU user sectors available. Cement chooses its second-lowest-cost option, CCS or CDR compensation, instead.

For Fig. 3c, compatibility with climate neutrality: Instead of the basic abatement cost, the extended abatement cost (Eq. (4)) is used, which accounts for the cost of compensation of residual emissions of the abatement options. Moreover, the full CDR compensation option for each sector is excluded. CDR is instead reserved for abating residual emissions from incomplete abatement options. The CCU coupling condition used in Fig. 3b is also applied here.

## Data availability

All data and assumptions used can be accessed on the code repository (see the "Code availability" section) and are explained in the Supplementary Information file. An additional Zenodo repository contains the plotting data for Figs. 2–6 in an accessible format: https://doi.org/10.5281/zenodo.14382820 (ref. 85). Furthermore, the techno-economic analysis carried out in this study can be reproduced with adjusted assumptions using the interactive webapp at https://doi.org/10.5880/pik.2025.001 (ref. 87), indicating the individual cost components in the levelized cost calculation.

## Code availability

A permanent copy of the software used is available on Zenodo at https://doi.org/10.5281/zenodo.14284322 (ref. 86). It can also be accessed directly on GitHub: https://github.com/clarabachorz/mapping-hte-sectors.

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

## Acknowledgements

We thank U. Ruth for his valuable input and suggestions in the initial stages of the paper. C.B. gratefully acknowledges support from the Bosch Research Foundation. P.C.V. and F.U. received support for the research for this work from the Kopernikus-Ariadne project by the German Federal Ministry of Education and Research (grant nos. 03SFK5A, 03SFK5A0-2) and the HyValue project (grant no. 333151). F.U. and L.G. received support for the research for this work from the CCfD project by the German Federal Ministry for Economic Affairs and Climate Action (grant no. DE64B88F-54CC-4CFB-B802-7CEC1AA2E542). This research also received funding from the European Union's Horizon Europe research and innovation program (PRISMA, grant no. 101081604).

## Author contributions

F.U. suggested the research question. C.B. and F.U. jointly conceived and designed the study in consultation with P.C.V. C.B. and P.C.V. curated the techno-economic data. C.B. wrote the software for performing the calculations and created the figures. C.B., F.U., and P.C.V. wrote the manuscript with contributions from G.L.

## Funding

## Competing interests

The authors declare no competing interests.
