## [Transparent Peer Review file · Nature Communications]

Exploring techno-economic landscapes of abatement options for hard-to-electrify sectors

Corresponding Author: Ms Clara Bachorz

This manuscript has been previously reviewed at another journal that is not operating a transparent peer review scheme. The manuscript was considered suitable for publication without further review at Nature Communications.

Version 0:

Reviewer comments:

Reviewer #1

(Remarks to the Author)
Dear Authros,

I was sceptical at first but I think you have made substantial changes that better motivates the paper and thus adress my main concerns. It is now more clear what your contribution is. It is still a result built on many many unreliable future cost assumptions but at least you can conceptually frame the choices in what you call HTE sectors with this paper. I appreciate the efforts made to clarify the objectives and the framing of the paper.

Comments: Could you use the same metric for hydrogen cost ? Now you shift between Euros/Kg and Euros/MWh ? (most publications use euros/kg today but that is not necessary)

Page 5: "decentralised sites like paper and pulp...." Not sure I would claim that paper and pulp facilities are "decentralised" the same way as e.g. waste combustion

Fig2. Maybe difficult to change but I note that you show varying hydrogen cost but assume a steady BF-BFO-CCS cost - the cost of BF-BOF-CCS is, as you state in the discussion, very uncertain

page 15 BF-BOF Discounted BF-BOFs ? A BF-BOF is relined every 18-23 years (roughly) for a cost representing roughly 50% of greenfield capex. Is that accounted for or are you "within the 20 years"?

Limitatin and furhter work. 4th line. A reference is missing

(Remarks on code availability)

Reviewer #3

(Remarks to the Author)
I would like to thank the authors for the revisions to their manuscript. I believe it is now ready for publication.

(Remarks on code availability)

Version 1:

Reviewer comments:

Reviewer #1

(Remarks to the Author)

Dear Authoes,

You have adressed all my concerns. I think it is ready for publication now

(Remarks on code availability)

General response to the reviewers

We thank the reviewers for their further efforts and feedback. We have incorporated all their suggestions to the revised manuscript (see point-per-point response below). The majority of changes to the manuscript are formatting edits, slight reformulations and a minor color palette change. Figure 2 now includes an uncertainty range on the BF-BOF-CCS option's final levelized cost and FSCP, as a result of varying the CAPEX of implementing carbon capture.

Reviewer-specific responses

Reviewer #1

Dear Authors,

I was sceptical at first but I think you have made substantial changes that better motivates the paper and thus address my main concerns. It is now more clear what your contribution is. It is still a result built on many many unreliable future cost assumptions but at least you can conceptually frame the choices in what you call HTE sectors with this paper. I appreciate the efforts made to clarify the objectives and the framing of the paper.

- We thank the reviewer for their comment. We are glad that we were able to address your main concerns, and believe your comments have helped significantly improve the manuscript.

Comments: Could you use the same metric for hydrogen cost ? Now you shift between Euros/Kg and Euros/MWh ? (most publications use euros/kg today but that is not necessary)

- We thank the reviewer for pointing out the inconsistency in notation. We have modified it and now consistently use €/MWh, and additionally specify the converted equivalent in €/kg in brackets, for the conclusion.

Page 5: "decentralised sites like paper and pulp...." Not sure I would claim that paper and pulp facilities are "decentralised" the same way as e.g. waste combustion

- We thank the reviewer for catching this inaccuracy. It was corrected in the manuscript: *"Biogenic CO₂ could be a cheaper option; however, it requires capturing waste CO₂ from decentralized sites ~~suchlike as pulp and paper production facilities,~~ waste or biogas plants, or from pulp and paper production facilities, for which there are currently no economic incentives."*

Fig2. Maybe difficult to change but I note that you show varying hydrogen cost but assume a steady BF-BFO-CCS cost - the cost of BF-BOF-CCS is, as you state in the discussion, very uncertain

- We thank the reviewer for their suggestion. We have implemented it, and slightly appended the text to reflect this change. Doing this calculation has helped us realise that while the relative uncertainty on the CAPEX of CCS to a BF-BOF is high, the impact on overall levelized cost and abatement cost is small. We have therefore updated our discussion concerning BF-BOF-CCS, changing our wording from "big source of uncertainty" to "source of uncertainty". The updated Figure 2 can be found below:

Figure 2. Cost breakdown of abatement options for steel for different hydrogen cost assumptions (new plant). **Panel (a)** shows the levelized cost breakdown for the fossil steel reference, a blast furnace with a basic oxygen furnace (BF-BOF), compared with two greener alternatives, BF-BOF-CCS (BF-BOF with carbon capture and storage) and H2-DRI-EAF (direct reduction of iron with an electric arc furnace operating with low-emission hydrogen). The levelized cost breakdown of the H2-DRI-EAF option is shown under four assumptions for the cost of low-emission hydrogen (H2). The CAPEX for carbon capture on a BF-BOF is also varied by $\pm 50\%$, as indicated by the error bar (SD). **Panel (b)** shows the calculated fuel-switching CO₂ price (FSCP) for both options, as a function of the hydrogen cost assumed. The blue ribbon indicates the uncertainty on the FSCP of BF-BOF-CCS, based on varying the BF-BOF carbon capture CAPEX. CAPEX: capital expenditure, OPEX: operational expenditure, O&M: operation and maintenance costs, PCI: pulverized coal injection.

page 15 BF-BOF Discounted BF-BOFs ? A BF-BOF is relined every 18-23 years (roughly) for a cost representing roughly 50% of greenfield capex. Is that accounted for or are you "within the 20 years"?

- We thank the reviewer for raising this clarification point. When changing the fossil reference to be an already existing BF-BOF plant, we indeed assume that we are "within the 20 years", more accurately, we assume that a relining has just been carried out and the associated costs are thus sunk.

We clarified this point in the manuscript:

" To evaluate this, we study the case of an operating BF-BOF steel plant, by using a reference BF-BOF plant that has already been fully depreciated (no capital expenditure in the levelized cost components) - this includes the assumption that a blast furnace relining has just been carried out (considering a typical relining cycle has a duration of 15-20 years)."

Limitatin and furhter work. 4th line. A reference is missing

- We thank the reviewer for catching this error. It was corrected.

Reviewer #3 (Remarks to the Author):

I would like to thank the authors for the revisions to their manuscript. I believe it is now ready for publication.

- We thank the reviewer for their input, and help in improving this manuscript.